

# The extent of carbapenemase-encoding genes in public genome sequences

Ingmar Janse[1,*], Rick Beeloo[1,*], Arno Swart[1], Michael Visser[2], Leo Schouls[1], Engeline van Duijkeren[1] and Mark W.J. van Passel[1,3]

[1] Center for Infectious Disease Control, National Institute for Public Health and the Environment, Bilthoven, Utrecht, The Netherlands
[2] Sequencing and Bioinformatics, Netherlands Food and Consumer Product Safety Authority (NVWA), Utrecht, The Netherlands
[3] Ministry of Health, Welfare and Sport, The Hague, The Netherlands
[*] These authors contributed equally to this work.

Corresponding author
Ingmar Janse, ingmar.janse@rivm.nl

## ABSTRACT

Genome sequences provide information on the genetic elements present in an organism, and currently there are databases containing hundreds of thousands of bacterial genome sequences. These repositories allow for mining patterns concerning antibiotic resistance gene occurrence in both pathogenic and non-pathogenic bacteria in e.g. natural or animal environments, and link these to relevant metadata such as bacterial host species, country and year of isolation, and co-occurrence with other resistance genes. In addition, the advances in the prediction of mobile genetic elements, and discerning chromosomal from plasmid DNA, broadens our view on the mechanism mediating dissemination. In this study we utilize the vast amount of data in the public database PATRIC to investigate the dissemination of carbapenemase-encoding genes (CEGs), the emergence and spread of which is considered a grave public health concern. Based on publicly available genome sequences from PATRIC and manually curated CEG sequences from the beta lactam database, we found 7,964 bacterial genomes, belonging to at least 70 distinct species, that carry in total 9,892 CEGs, amongst which $bla_{NDM}$, $bla_{OXA}$, $bla_{VIM}$, $bla_{IMP}$ and $bla_{KPC}$. We were able to distinguish between chromosomally located resistance genes (4,137; 42%) and plasmid-located resistance genes (5,753; 58%). We found that a large proportion of the identified CEGs were identical, i.e. displayed 100% nucleotide similarity in multiple bacterial species (8,361 out of 9,892 genes; 85%). For example, the New Delhi metallo-beta-lactamase NDM-1 was found in 42 distinct bacterial species, and present in seven different environments. Our data show the extent of carbapenem-resistance far beyond the canonical species *Acetinobacter baumannii, Klebsiella pneumoniae* or *Pseudomonas aeruginosa*. These types of data complement previous systematic reviews, in which carbapenem-resistant Enterobacteriaceae were found in wildlife, livestock and companion animals. Considering the widespread distribution of CEGs, we see a need for comprehensive surveillance and transmission studies covering more host species and environments, akin to previous extensive surveys that focused on extended spectrum beta-lactamases. This may help to fully appreciate the spread of CEGs and improve the understanding of mechanisms underlying transmission, which could lead to interventions minimizing transmission to humans.

## INTRODUCTION

Twenty-five years since the sequencing revolution started we see the availability of hundreds of thousands of bacterial genomes (*Wattam et al., 2017*), enabling a vast number of comparative analyses leading to e.g., virulence factor prediction (*Mao et al., 2015*) or metabolic pathway modelling (*Karp et al., 2010*). The extensive genome resource PATRIC (*Wattam et al., 2017*) includes detailed information on resistance genes (*Antonopoulos et al., 2019*), and amongst the many different uses for these accumulated data, they also aim at predicting antimicrobial resistance (AMR) phenotypes in individual genomes. However, the accumulation of all genetic data with their associated metadata on bacterial host species, isolation source, year of isolation, and co-occurrence with other clinical resistance genes, also allows for a global view into the dissemination of resistance genes, for example of carbapenemase-encoding genes (CEGs) in high priority pathogens (*Rello et al., 2019*).

Recently, *Kock et al. (2018)* performed a systematic literature review on the occurrence of carbapenem-resistant Enterobacteriaceae (CRE). The authors concluded that the prevalence of CRE in wildlife, livestock, companion animals and directly exposed humans poses public health risks, which was further corroborated by a study on the worsening epidemiological situation in Europe of carbapenemase-producing Enterobacteriaceae (*Brolund et al., 2019*). The literature review by Köck et al. was limited to Enterobacteriaceae, and environmental prevalence of CRE had not been included, even though genes encoding most acquired carbapenemases are believed to have transferred from environmental bacteria into species with clinical relevance (*Woodford et al., 2014*). We propose that evaluating carbapenemase-encoding genes and their associated metadata from genomic databases may add to the broader picture of the distribution of these genes throughout relevant (non-human) environments, and may highlight transmission between bacterial species via mobile genetic elements. These databases can show whether other bacterial species can represent CEG reservoirs, even when the original researchers of these genomes may not have annotated the carbapenem-resistance potential of these species. We aim to tabulate carbapenemase-encoding genes from an extensive genome database, in order to appraise their distribution patterns. Our objective was not a comprehensive coverage of all CEG variants, instead we chose a conservative approach for the annotation of CEGs and used identical sequences to highlight their distribution and mobility.

## MATERIALS & METHODS

All available 175,882 bacterial genomes (August 2018) and their corresponding metadata were downloaded from PATRIC (*Wattam et al., 2017*). This metadata included information on the isolated bacteria such as species, taxonomic lineage, isolation source (host organisms, e.g. human, or environment, e.g. forest or wastewater). In addition, the

nucleotide (.ffn) gene and protein sequences (.faa) were retrieved from the FTP server (http://ftp.patricbrc.org). Because genome information is recorded in free-text fields which hampers unambiguous analysis, taxonomic lineages for each genome were re-annotated at the species and genus level. Species names were harmonized by automatic annotation as well as manual curation. In order to bin isolation source metadata from all genomes correctly, the various formats describing hosts from which bacteria had been isolated were queried against the NCBI taxonomy database to retrieve taxonomic lineages. Standardized host names allowed assignment to groups such as cattle, chicken, and human. Source metadata from environmental or food isolates were standardized by querying these against manually curated keywords (File S1) to assign each genome to specified environments (File S2). In case source metadata was absent, the genome was designated ''miscellaneous''.

To create a reliable antibiotic resistance gene (ARG) set, protein sequences of all PATRIC ARGs identified upon comparison with CARD (indicated by the ''source'' column in PATRIC annotation) were subjected to the Resistance Gene Identifier (RGI) (*Jia et al., 2017*) and only those detected by the protein homolog model with a perfect or strict cut-off were kept. To identify carbapenemase-encoding genes in this dataset, the protein sequences of 566 manually curated CEGs (File S3) were obtained from the Beta-lactamase database (BLDB) (*Naas et al., 2017*) as it was compiled in August 2018 and detected in the ARG set using TBLASTN (100% identity and coverage). Although less stringent settings would have identified more potential CEGs, we chose a conservative approach in order to avoid erroneous assignments. The CEGs were clustered on 100% sequence identity and coverage using Python, and their genomic locations (chromosome or plasmid) were predicted using MOB-RECON (*Robertson & Nash, 2018*).

## RESULTS

### Carbapenemase-encoding genes in bacterial genomes

After RGI verification of annotated antibiotic resistance genes from PATRIC, we obtained nearly 2 million ARGs from over 170,000 bacterial genomes (covering almost 11,000 species), from which set we identified 9,892 carbapenemase-encoding genes in 7,964 bacterial genomes. The set of 9,892 CEGs consisted of 203 unique protein sequences and 257 unique nucleotide sequences. The former reflect the amino-acid based beta-lactam classifications system used by the BLDB (*Naas et al., 2017*) that allowed accurate assignment of each CEG to its protein name. Genes encoding KPC-2, OXA-23, OXA-66, and KPC-3 were detected over 1,000 times in our dataset. The 7,964 genomes belonged to at least 77 bacterial species (18 lacked taxonomic assignment beyond the genus level, File S4), from 33 genera, 18 families and 5 phyla.

### Multiple occurrences of CEGs within a single genome

We found that while the majority (6,191) of genomes only encoded a single CEG, 1,773 genomes carried multiple CEGs. Of the genomes carrying multiple CEGs, 189 harboured CEGs encoding the same protein, with OXA-23 encoding genes being most frequently present in multiple copies (41 genomes), followed by genes encoding KPC-3 ($n = 35$), KPC-2 ($n = 30$), NDM-1 ($n = 12$), and OXA-72 ($n = 9$). Interestingly, we found 38

genomes containing three or more genes encoding the same protein, and three genomes with five copies for the same protein, KPC-2 for *Klebsiella michiganensis*, and OXA-23 and OXA-72 for *Acinetobacter baumannii*. Moreover, we also found multiple gene copies encoding for the same protein in less common pathogens, such as three copies encoding KPC-3 in *Serratia marcescens* and *Raoultella planticola*.

Most of the genomes with multiple CEGs carried genes encoding different proteins ($n = 1,666$), which were of the same carbapenemase type for the majority of these genomes ($n = 1,533$). Combinations of OXA (oxacillinase) type CEGs were most common, especially $bla_{OXA-23}$ and $bla_{OXA-66}$ (803 genomes) and $bla_{OXA-23}$ and $bla_{OXA-82}$ (290 genomes). However, 133 genomes contained CEGs encoding carbapenemases of two different types. The most frequent combination of genes encoding distinct carbapenemase types, i.e., $bla_{NDM-1}$ and $bla_{OXA-232}$ were found in 25 *Klebsiella pneumoniae* genomes and one *Escherichia coli* genome. Other common combinations were $bla_{NDM-1}$ and $bla_{OXA-94}$ in 11 *Acinetobacter baumannii* genomes, $bla_{KPC-2}$ and $bla_{NDM-1}$ in five *Enterobacter cloacae*, two *Klebsiella pneumoniae*, two *Klebsiella michiganensis*, and one *Citrobacter freundii*, and lastly $bla_{NDM-1}$ and $bla_{OXA-23}$ in 10 *Acinetobacter baumannii* genomes, one *Acinetobacter radioresistens* genome and one *Acinetobacter* sp. genome.

## Distribution of CEGs between species and environments

Most CEGs were found in the genera *Acinetobacter* (3074 genomes: 4783 CEGs), *Klebsiella* (3267: 3404), *Escherichia* (542: 559), *Enterobacter* (485: 502) and *Pseudomonas* (268: 281). Overall, CEGs were found most frequently present in the species *Acinetobacter baumannii* (4,502 CEGs), *Klebsiella pneumoniae* (3,285 CEGs), *Escherichia coli* (557 CEGs), *Enterobacter cloacae* (313), *Pseudomonas aeruginosa* (245 CEGs), and also, albeit to lesser extent, in species less commonly included in carbapenemase surveillance such as *Serratia marcescens* (66 CEGs), *Citrobacter freundii* (59 CEGs), *Klebsiella oxytoca* (36), *Klebsiella quasipneumoniae* (27), *Klebsiella michiganensis* (24 CEGs), and *Proteus mirabilis* (24 CEGs), and *Acinetobacter radioresistens* (20 CEGs). File S5 presents the numbers of genomes and CEGs. The important CEG $bla_{IMP-1}$ was most often detected in *Enterobacter hormaechei* and *Klebsiella pneumoniae*, but also in *Acinetobacter baumannii*, *A. junii*, *A. pittii*, *A. nosocomialis*, *Enterobacter cloacae*, *E. kobei*, *E. asburiae*, *Escherichia coli*, *Pseudomonas aeruginosa* and *Serratia marcescens*. $bla_{OXA48}$ was predominantly found in *K. pneumoniae*, but also in *Escherichia coli*, *Enterobacter aerogenes*, *E. cloacae*, *E. hormaechei*, *E. kobei*, *Citrobacter freundii*, *C. koseri*, *Shewanella* sp. (on chromosome), *Kluyvera ascorbata*, *Proteus mirabilis* and *Raoultella ornithinolytica*. $bla_{NDM-1}$, $bla_{KPC-2}$ and $bla_{KPC-3}$ were most often identified in *K. pneumoniae*, followed by *E. coli* and *E. cloacae* (Files S6–S8).

Nucleotide sequence comparison of the 9,892 CEGs revealed that 1,591 CEG sequences were present in only a single bacterial species, whereas the remaining 8,301 CEG sequences were found identically in multiple species. For example, the well-known CEGs $bla_{NDM-1}$, $bla_{KPC-2}$, $bla_{IMP-1}$ and $bla_{OXA-48}$ in at least 38, 27, 12 and 13 species, respectively.

Based on the isolation information for each genome we distinguished 11 different environments of CEGs, of which the human isolates represented by far the largest
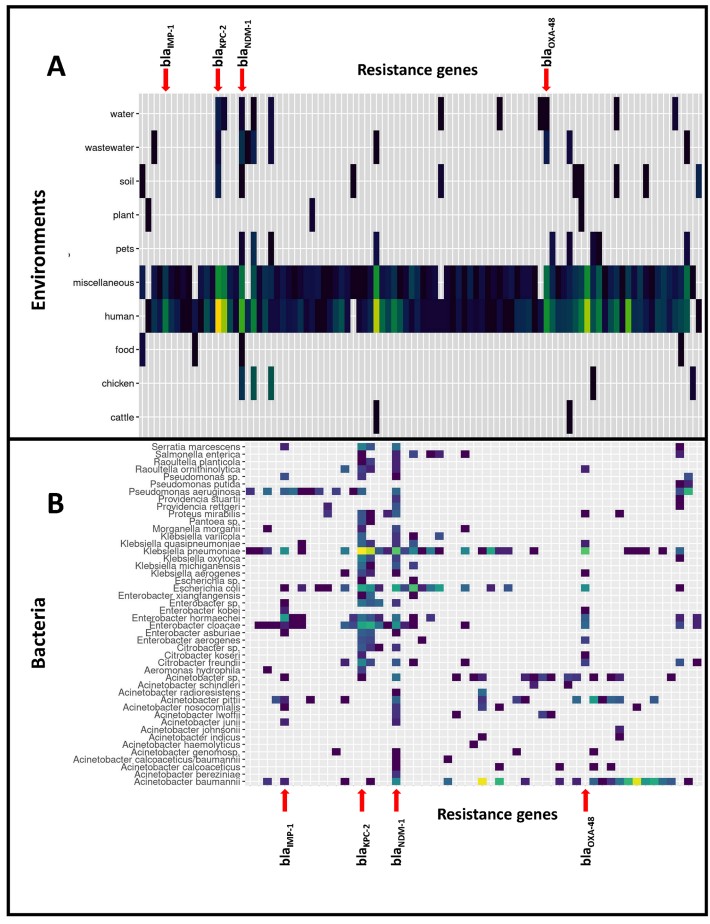

**Figure 1 Distribution of the 9,892 carbapenemase-encoding genes in bacterial species and non-human reservoirs.** Horizontal lines represent reservoirs (A) or bacterial species (B) that collectively harbor different CEGs. Vertical lines represent CEGs that occur in many different bacterial species or reservoirs: four genes have been highlighted. The figure was simplified by removing the rare CEGs, and the full data is available in Files S6–S8.

environment in this study (8,099 CEGs, 82%). CEGs such as $bla_{KPC-2}$, $bla_{OXA-23}$ and $bla_{OXA-66}$ were dominant when considering all environments. However, when the human and miscellaneous (which will likely contain many human isolates) genomes were excluded from the dataset, dominant CEGs shifted, leading to high frequencies of $bla_{NDM-5}$, $bla_{NDM-9}$ and $bla_{NDM-1}$, mostly identified in bacteria isolated from chickens (109 genomes). The distribution of CEGs over the different species and environments is presented in Fig. 1.

## Genomic location of CEGs

We evaluated the genomic location of CEGs based on their genomic context and found that 4,137 (42%) were predicted to be chromosomally located while 5,753 (58%) were predicted to be plasmid-borne $bla_{IMP1}$ was most often located on IncN or IncW plasmids, $bla_{KPC-2}$ on IncFII or IncU plasmids, $bla_{KPC-3}$ on IncFIA or IncFII plasmids, $bla_{NDM-1}$ on IncA/C plasmids, $bla_{NDM-5}$ on IncX3 plasmids and $bla_{OXA-48}$ on Inc L/M plasmids.

**Table 1  Distribution of carbapenemase-encoding genes (CEG) in bacterial genomes.** The 9,892 CEGs occur in 7,884 bacterial genomes. The majority of genes (6,191) is present in a single copy in its genome, but 3,701 genes reside in a genome with additional carbapenemase-encoding genes.

| | | Number of different CEGs | | |
| --- | --- | --- | --- | --- |
| | | 1 | 2 | 3 |
| Number of CEGs per genome | 1 | *6,191* | 0 | 0 |
| | 2 | 81 | *1,567* | 0 |
| | 3 | 21 | 52 | *32* |
| | 4 | 4 | 8 | 0 |
| | 5 | 1 | 4 | 1 |
| | 6 | 0 | 2 | 0 |

## DISCUSSION

The identification of CEGs in a large number of genomes of a diversity of bacterial species showed that the distribution of these high-priority resistance genes extends far beyond the well-known clinically relevant carbapenem-resistant *Acinetobacter baumannii*, *Pseudomonas aeruginosa* and *Enterobacteriaceae,* which are known to be emerging causes for hospital acquired infections (https://www.who.int/infection-prevention/publications/guidelines-cre/en/).

A recent cross-border collaborative effort on whole genome sequencing of certain carbapenemase positive *K. pneumoniae* identified muli-country transmission clusters (*Ludden et al., 2020*). This analysis relied on 143 genome sequences, and underlined the benefits of data sharing. The integration of public data sources, such as the PATRIC genome repository with a backdrop of over 3,200 carbapenemase containing *K. pneumoniae* genomes, in addition to a substantive potential reservoir of mobile carbapenemase-encoding genes, may provide further context and epidemiological links for both strains as well as mobile elements outside the regularly surveilled species.

A considerable number of these genomes carried two or more different CEGs (Table 1). Co-expression of different types of carbapenemase genes, for example $bla_{OXA-48}$ and $bla_{NDM-1}$, together with other resistance genes that might be present on the plasmids, limits treatment options  (*Gona et al., 2019*). In this study, only CEGs were included in the analyses, but their co-occurrence with antibiotic resistance genes from completely different classes, such as aminoglycosides, tetracyclines, et cetera, may be of interest as well to evaluate the emergence of multidrug resistant pathogens.

We found that identical CEGs are frequently shared between bacterial species and -as a consequence- can be encountered in a large number of environments (Fig. 1). Horizontal gene transfer is an important evolutionary force for adaptation, and mobile genetic elements are well-known vectors for gene exchange, especially for antibiotic resistance genes (*Ding et al., 2016*). The high percentage (58%) of predicted plasmid-located CEGs in the genomes indeed shows that a large proportion of CEGs have an enhanced capacity for horizontal gene transfer. Previous genomic epidemiology studies may not have fully appreciated the widespread character of CEG dissemination (*Wilson & Torok, 2018*), and possible co-mobility with other resistance determinants.

**Table 2  Diversity of CEGs in the 11 reservoirs distinguished in the genome database.**

| Environment | # Distinct CEGs | CEGs | # Species | Species[a] |
|---|---|---|---|---|
| Pig | 1 | NDM-1 | 1 | *A. baumannii* |
| Cattle | 3 | OXA-146,OXA-23,OXA-58 | 1 | *A. indicus* |
| Plant | 3 | CGB-1,OXA-120,OXA-65 | 2 | *A. baumannii, C. indologenes,* |
| Chicken | 5 | NDM-1,NDM-5,NDM-9,OXA-68,VIM-48 | 5 | *A. baumannii, E. coli, K. pneumoniae, S. enterica, P. putida* |
| Food | 5 | BcII-1,IMP-27,NDM-1,OXA-497,VIM-1 | 6 | *A. baumanni, B. cereus, B. thuringensis, E. coli, V. parahaemolyticus, S. enterica* |
| Pets | 9 | NDM-1,NDM-5,NDM-9,OXA-23,OXA-500,OXA-58,OXA-68,OXA-69,VIM-2 | 6 | *A. baumannii, A. gandensis, A. pittii, A. radioresistens, E. coli, P. aeruginosa* |
| Soil | 10 | BcII-1,KPC-2,NDM-1,OXA-213,OXA-273,OXA-64,OXA-65,OXA-72,OXA-91,VIM-5 | 10 | *A. baumannii, A. calcoaceticus, Acinetobacter sp., B. cereus, B. subtilis, C. braakii, C. freundii, Erythrobacter sp., P. guariconensis, P. plecoglossicida* |
| Wastewater | 11 | GES-5,KPC-2,NDM-1,NDM-4,NDM-5,NDM-9,OXA-23,OXA-48,OXA-58,SHV-38,VIM-2 | 11 | *A. johnsonii, A. junii, Acinetobacter sp., A. hydrophila, C. freundii, E. cloacae, E. kobei, E. coli, Escherichia sp., K. pneumoniae, Pseudomonas sp.* |
| Water | 13 | KPC-2,KPC-3,NDM-1,NDM-5,NDM-9,OXA-269,OXA-273,OXA-280,OXA-360,OXA-444,OXA-48,OXA-72,SPM-1 | 14 | *A. calcoaceticus, A. johnsonii, A. pittii, A. schindleri, E. cloacae, E. coli, G. pentaromativorans, K. pneumoniae, K. quasipneumoniae, K. variicola, P. aeruginosa, Pseudomonas sp., Ralstonia sp., V. cholerae* |
| Miscellaneous | 109 | [b] | 48 | [c] |
| Human | 175 | [b] | 73 | [c] |

**Notes.**
[a] For the full names, please refer to File S4.
[b] The list of genes is in the File S6.
[c] The list of species is in the File S7.

The database approach identified human isolates as the largest reservoir of CEGs. This however does not necessarily mean that the human environment is the dominant reservoir for carbapenem-resistance genes, but rather may reflect the frequent sampling of humans. In addition, it is likely that many more CEG-environment combinations exist as this analysis is limited to the biased isolates present in the database. Indeed, most non-human environments harbor multiple CEG variants (Table 2), despite the fact that they contain relatively few isolates.

## CONCLUSIONS

Large genome repositories are treasure troves for comparative analyses in different fields of study, even though these databases do not represent surveillance data and therefore contain bacterial isolates at other frequencies compared to their isolation sources, nor are they free of annotation errors. Still, owing to their vast sizes, genome sequence collections allow for an alternative view on the distribution of resistance genes and exchange potential between bacteria and environments. Carbapenemase-encoding genes are distributed over a large number of bacterial species, with some CEGs, such as $bla_{NDM-1}$, present in at least

42 different species, isolated from at least seven distinct non-human environments. These numbers show the breadth of reservoirs and the vast potential for resistance gene exchange and transmission. Data sharing and the use of open repositories for surveillance purposes would therefore benefit evaluations of health risks for resistant infections.

### Funding
The authors received no funding for this work.

### Competing Interests
The authors declare there are no competing interests.

### Author Contributions
- Ingmar Janse and Rick Beeloo conceived and designed the experiments, performed the experiments, analyzed the data, prepared figures and/or tables, authored or reviewed drafts of the paper, and approved the final draft.
- Arno Swart conceived and designed the experiments, performed the experiments, prepared figures and/or tables, authored or reviewed drafts of the paper, and approved the final draft.
- Michael Visser conceived and designed the experiments, performed the experiments, authored or reviewed drafts of the paper, and approved the final draft.
- Leo Schouls and Engeline van Duijkeren analyzed the data, authored or reviewed drafts of the paper, and approved the final draft.
- Mark W.J. van Passel conceived and designed the experiments, analyzed the data, prepared figures and/or tables, authored or reviewed drafts of the paper, and approved the final draft.

### Data Availability
All genome sequences and metadata are available at PATRIC: https://www.patricbrc.org/. Genomes were downloaded August 2018 and the list of IDs can be found at GitHub (https://github.com/rickbeeloo/PATRIC-carba).

The database was built and queried by using the R software environment (https://www.r-project.org/).

### Supplemental Information
Supplemental information for this article can be found online at http://dx.doi.org/10.7717/peerj.11000#supplemental-information.

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
