# Peer review of "The extent of carbapenemase-encoding genes in public genome sequences"

_PeerJ, doi:10.7717/peerj.11000_

## Round 0.1 · original submission · Major Revisions

Dear Dr. Janse and colleagues:

Thanks for submitting your manuscript to PeerJ. I have now received three independent reviews of your work, and as you will see, one reviewer recommended rejection, while the other two suggested minor revision (yet with many suggested changes). I am affording you the option of revising your manuscript according to all three reviews.

The greatest concern is the nature of the in silico work; you must be more thorough in your revision with describing the potential origins of carbapenemase-encoding genes. This will entail more rigorous bioinformatics analyses, as well as robust comparative genomic analysis where possible. Please also provide all relevant metadata for each sequence and analyzed genome, including whether or not genomes are closed or open (assembly with greater than one contig) and the nature of plasmids.

Reviewer 3 described a neat approach for considering the origins of carbapenemase-encoding genes.

Please be more thorough with description of your analyses. Ensure that all of the data is included and easy to obtain. Your work should be repeatable. A more inclusive background on carbapenemase-encoding genes should be provided in the Introduction, with relevant literature cited.

The reviewers raised many other concerns about the manuscript. Please address all of these issues.

I look forward to seeing your revision, and thanks again for submitting your work to PeerJ.

Good luck with your revision,

-joe

Reviewer 1 ·

Basic reporting

This is a nice and clearly written manuscript by Janse et al who have searched for carpapenemase–encoding genes in the public database PATRIC. Overall, the structure and language of the manuscript are good and I only have some minor comments.

1. In line 149, you write “…Acinetobacteria (3074 genomes: 4783 genes), here it is unclear if you mean that all genomes belonging to the genus Acinetobacteria contained a total of 4783 genes or if you have discovered 4783 CEGs in these genomes. This is repeated a couple of times throughout the manuscript (line 165, 166 and 170 for instance). The authors need to be consistent with the nomenclature, if you mean CEGs, please write so, and if you mean genes, please clarify.

2. Line 90-92: I had trouble understanding the purpose of this sentence (and approach). Firstly, it becomes unclear when using the word “host” in this context, since when searching for ARGs a host is mostly considered the bacteria carrying the ARG. Secondly, why is the taxonomic lineage of the isolation source needed when you have the keywords to classify the sources? I can’t find any part in the manuscript where this information is used. These lines need to be clarified.

3. As there often are many duplicates in genomic databases and, especially, the genomes of some species make up big parts of these databases, I miss a number stating the taxonomic diversity of the analyzed data. Since this most definitely will reflect on the number of species you will find CEGs in, it is important to know how many species that were present in the data to begin with.

4. In the abstract you write “...antibiotic resistance gene occurrence in (pathogenic) bacteria”. However, you have searched all genomes in the PATRIC database, and it is my understanding that this database not only includes pathogenic bacteria. I therefore recommend to either clarify this sentence or remove the word pathogenic.

5. Line 118. It is stated that “from the collection of 2 million antibiotic resistance genes in over 170 000 bacterial genomes…”. Here I wonder what the collection is? Is it that it was 2 million genes annotated as ARGs by PATRICs automatic annotation or is it based on the number you got after you had extracted those annotated as ARGs by PATRIC and then analyzed them using RGI?

6. Line 86-88: The information about how the suffixes “sp.” and “subsp.” were removed is somewhat trivial information that rather complicated the reading than giving important information about how the study was conducted. I suggest removing this sentence.

7. Line 88: The sentence about the free text field seems misplaced. Please elaborate or remove it.

8. Lines 143-147: Please spell out numbers lower than 11 (i.e. write “one” instead of 1).

9. The abbreviation CEG is introduced in every section except in the introduction where you use both carbapenemase-encoding genes and CEGs. Please go through the article so that it is consistent with the introduction and use of CEG. Also, sometimes you spell it “carbapenemase-encoding” and sometimes you omit the hyphen.

10. In table 2 you list three footnotes but I can only see two referenced in the table (* and ***).

Experimental design

1. The authors have identified CEGs in the genomic data by first extracting all protein sequences annotated as ARGs in the PATRIC database, then these potential ARGS were analyzed using Resistance Gene Identifier. Those that passed the strict-cutoff by RGI were then subjected to a blast search against a manually collected set of carbapenemases-encoding genes from the beta-lactamase database. The analysis is thoroughly done and I have confidence in the validity of the reported numbers. However, I have some questions regarding the identification of CEGs.

a) Why did you not search either all genomes or protein sequences in PATRIC for CEGs? I.e. what was the purpose of going through the two intermediate steps of extracting only ARGs as annotated by PATRIC and then RGI? Since you in the end used a TBLASTN against the, from the beta-lactamase database, manually collected set of CEGs with 100% identity and coverage threshold, you can be very certain you don’t get any false positive. Instead, you risk missing true positives by going through the first two steps (PATRIC and RGI).

b) Since you are using a very strict cut-off, you will only find exactly what you are searching for (i.e. those exact genes you have included in your set of reference CEGs), and there’s a big risk of failing to detect many CEGs that have not yet been included in the database. This needs to be acknowledged. I would furthermore like to know what date these CEGs were collected from the beta-lactamase database as the number of validated CEGs constantly is increasing, and as mentioned, you will most likely only find the CEGs that are included in your set.

c) It would be interesting to know the class distribution of the CEGs included in your manually collected set (i.e. how many class A, B, C and D), and the class distribution of the ones identified in this study.

2. In line 148 it is stated that most CEGs were found in the genera of Acinetobacter. Although the absolute number of detected CEGs is the highest in Acinetobacter, it doesn’t necessarily mean that they are the most likely carrier of CEGs but could instead be a reflection of what’s in the database. It would be interesting if the authors could put these numbers in perceptive by comparing them to the number of Acinetobacter, Klebsiella etc. present in the database. I interpret the “3074 genomes: 4783 genes” as that the authors detected 4783 CEGs in 3074 Acinetobacter genomes, but then there’s no information on how many Acinteobacter genomes there originally were in the database.

Validity of the findings

1. The authors state on line 221 that “the database approach identified human isolates as the largest reservoir of CEGs”. However, the number of identified CEGs in different environments will largely reflect the composition of the database. As the authors mention in line 222-223, since the database most likely consist of many genomes isolated from humans, it is not strange that you also find many CEGs there. Furthermore, one must also consider that there is a bias in the sequencing where pathogenic bacteria potentially carrying CEGs could be more abundant in the database. However, the study would benefit from actually checking if the proportion of CEGs identified in bacteria from human isolates was significantly higher than the proportion of CEGs identified in bacteria from other sources (this can be tested with for instance Fisher’s exact test). Otherwise, I suggest a rephrasing of the sentence on line 221.

2. In line 216 it is stated that a large proportion of CEGs have an enhanced capacity for horizontal gene transfer. This statement is based on that 58% of the CEGs detected by the authors were predicted to be plasmid located. However, one has to be careful about drawing conclusions about the real proportion of CEGs that have the potential for horizontal gene transfer. Firstly, the authors have only searched for a limited set of CEGs, using a strict cutoff, where many of these are already confirmed mobile CEGs. Therefore it also likely that many of the, in this study identified, CEGs will be mobile. Secondly, the database is likely biased towards human isolates (See above comment) and there we will probably also find many of the validated mobile CEGs. The authors should therefore rewrite this paragraph.

3. Line 225. The last sentence here seems a bit out of context. Do the non-human reservoirs have more genomes that contain more than one type of CEG compared to the human reservoir? And if so, what would that implicate? This result is not presented in the main text and Table 2 as the authors refer to here does not contain any information about the reservoirs.

Reviewer 2 ·

Basic reporting

This manuscript failed in reporting clear and unambiguous data. Indeed, there is lots of confusion throughout since the authors seem not to possess a sufficient knowledge in carbapenemases. A significant confusion comes from the lack of differentiation between the intrinsic genes (or naturally-occurring in certain species), and the acquired genes. Without addressing clearly that differentiation, and allaying data accordingly, the whole manuscript is meaningless.
Some examples; OXA-66-like enzymes are ALWAYS present in Acinetobacter baumannii. Thus not surprising to see an association with OXA-23 when that latter is acquired !

Experimental design

Unclear whether only whole genome sequences deposited have been retained, or the analysis actually included also incomplete genomes ?
What was the cut-off value retained in term of sequence identity to retain some sequences and not others when variants (close or more distant) are identified ?

Validity of the findings

The validity of data cannot be assessed, since this analysis is a pure in-silico analysis. Therefore, whether a given gene was indeed identified in a given bacterial species remains always doubtful. To the experience of the reviewer, it is very common to have genomic data deposited with inaccurate bacterial species identification. This makes the whole analysis doubtful very often.

Additional comments

Poor manuscript resulting from a poor analysis of in-silico data, without corresponding basic knowledge in the field.

·

Basic reporting

No comment

Experimental design

The paper describes the distribution of carbapenemases by host species and by environment and collates valuable data. The authors also predict whether genes are chromosomal or plasmid borne. However I couldn't really see the analysis or the results reported regarding chromosomal contexts of the important clinical genes that are plasmid borne and responsible for resistance in key pathogens. Could more information be extracted on likely origins of these genes such as NDM1. Which species carry these genes on their chromosomes, and are they associated with MGEs on the chromosome. Can the authors identify likely original hosts of genes?

In addition if a lower cutoff was used the authors might identify divergent gens that have not already been identified and this might also give information on progenitor genes and original hosts of these genes. Perhaps this could be done just for some of the NDM genes for example?

Validity of the findings

More could be made of the differentiation between plasmid and chromosomal location, particularly if chromosomal location without MGEs such as integrons, transposons, IS elements can be determiend.

Additional comments

This is an interesting paper but I was disappointed that analysis and discussion of the origins of these genes was not developed further given the amount of information available. Some further analysis focusing on chromosomal contexts (ie. potential origins) and also of divergent genes (85-90% similarity?) could reveal extremely important insights into the origins, evolution and transfer of clinically important carbapenemases. This would dramatically increase the significance of this study.

---

## Round 0.2 · accepted · Accept

Dear Dr. Janse and colleagues:

Thanks for revising your manuscript based on the concerns raised by the reviewers. I now believe that your manuscript is suitable for publication. Congratulations! I look forward to seeing this work in print, and I anticipate it being an important resource for groups studying carbapenemases. Thanks again for choosing PeerJ to publish such important work.

Best,

-joe

Reviewer 1 ·

Basic reporting

No comment

Experimental design

No comment

Validity of the findings

No comment

Reviewer 2 ·

Basic reporting

The revised version has been improved with respect to the original version, but is still of very limited interest according to the referee's perspective.

Experimental design

Seems appropriate, even we must highlight here the strong bias created by the selection of complete genomes. Indeed, when it is said for instance that Klebsiella and Acinetobacter are the genera in which the highest number of carbapenemase genes were identified, this is basically resulting from a higher number of sequenced genomes I guess. Indeed, these are the genera for which we do encounter multidrug resistant isolates most often, and researchers are consequently focusing on those. Thus the distribution issue as presented in a dedicated paragraph is of limited interest.

Validity of the findings

Data are likely valid, and the analysis properly performed.

Additional comments

Still the same feeling I had with the former version. Of extremely limited interest.